# Mitochondria in Diabetic Kidney Disease

**DOI:** 10.3390/cells10112945

**Published:** 2021-10-29

**Authors:** Amna Ayesha Ahmad, Shayna Odeal Draves, Mariana Rosca

**Affiliations:** Department of Foundational Sciences, College of Medicine, Central Michigan University, Mount Pleasant, MI 48858, USA; ahmad2aa@cmich.edu (A.A.A.); drave1so@cmich.edu (S.O.D.)

**Keywords:** diabetes, diabetic kidney disease, mitochondria, bioenergetics, redox status

## Abstract

Diabetic kidney disease (DKD) is the leading cause of end stage renal disease (ESRD) in the USA. The pathogenesis of DKD is multifactorial and involves activation of multiple signaling pathways with merging outcomes including thickening of the basement membrane, podocyte loss, mesangial expansion, tubular atrophy, and interstitial inflammation and fibrosis. The glomerulo-tubular balance and tubule-glomerular feedback support an increased glomerular filtration and tubular reabsorption, with the latter relying heavily on ATP and increasing the energy demand. There is evidence that alterations in mitochondrial bioenergetics in kidney cells lead to these pathologic changes and contribute to the progression of DKD towards ESRD. This review will focus on the dialogue between alterations in bioenergetics in glomerular and tubular cells and its role in the development of DKD. Alterations in energy substrate selection, electron transport chain, ATP generation, oxidative stress, redox status, protein posttranslational modifications, mitochondrial dynamics, and quality control will be discussed. Understanding the role of bioenergetics in the progression of diabetic DKD may provide novel therapeutic approaches to delay its progression to ESRD.

## 1. Diabetic Kidney Disease: Generalities

Diabetic kidney disease (DKD) is one of the most common microvascular complications in diabetic patients and the leading cause of end stage renal disease (ESRD) in the USA [1] Approximately 30% of patients with Type 1 Diabetes (T1D) and 40% of patients with Type 2 Diabetes (T2D) progress to DKD. DKD is usually associated with other diabetes-related comorbidities including retinopathy and cardiovascular complications. The prevalence of DKD is rising due to the diabetes global pandemic and causes heavy social and economic burdens in many countries.

A urinary albumin to creatine ratio (ACR) higher than 30 mg/g [2] used to be the major DKD diagnostic criterion, and albuminuria was considered the single best predictor of kidney function deterioration. The albuminuria-focused model of DKD progression implies that mitigating albuminuria in diabetic patients would result in a better renal outcome. However, DKD progression to ESRD may follow two distinct pathways, with and without albuminuria, with the latter being more frequent in T2D [3]. Therefore, DKD is currently diagnosed by the progressive decrease in the estimated glomerular filtration rate (eGFR) below 60 mL/min/1.73 m^2^. The use of eGFR to diagnose DKD is based on epidemiological observations indicating that albuminuria may be reversible whereas eGFR continues to progress to ESRD [4]. However, in patients with low eGFR, the level of albuminuria remains an independent predictor of the decline in eGFR [5]. Therefore, albuminuria may be used as a surrogate in clinical trials aimed to determine the benefit of therapeutic strategies to delay DKD progression [6]. Current guidelines recommend to assess both albuminuria and eGFR for DKD screening [7].

Early diagnosis is critical in preventing the DKD progression. It is recommended that T1D patients receive annual screening starting 5 years after the DKD diagnosis, while T2D patients must be screened at the time of diagnosis. In T1D, DKD progresses from an early glomerular hyperfiltration with microalbuminuria (ACR of 30–300 mg/g) to macroalbuminuria (ACR > 300 mg/g) while kidney function declines to ESRD. In T2D, the onset is harder to determine, and albuminuria associated with DKD may occur before T2D is diagnosed. It is also reported that 25% of patients with T2D have little or no albuminuria while presenting renal structural alterations characteristic for DKD [8].

Diabetic kidney disease is considered a microvascular complication of diabetes and involves an increased endothelial permeability and dysregulation of the vascular tone not only in the glomeruli but also in the systemic circulation. Therefore, albuminuria reveals a systemic microvascular disease and represents a risk factor for cardiovascular complications [9,10]. Delaying the progression of DKD may reduce cardiovascular mortality. Different therapeutic strategies have been proposed to decrease DKD progression and are based mostly on diabetes care and reducing intraglomerular pressure [11] Despite the progress in therapeutic approaches, diabetes represents 50% of causes for ESRD [12], and DKD doubles the risk for all-cause diabetes-related mortality [13]. There is a high need for novel therapeutic targets to stop, delay, and even reverse kidney damage in diabetes.

## 2. Diabetic Kidney Disease: Pathogenesis

In the diabetic kidney, the glomerular hypertension and hyperfiltration with secondary albuminuria, thickening of the glomerular and tubular basement membranes, mesangial expansion, and podocyte loss are associated with glomerular sclerosis, tubular atrophy, and a progressively reduced kidney function with decreased eGFR [1].

Macroscopically, diabetic kidneys are larger than normal due to expansion of the nephron, particularly mesangial enlargement and proximal tubule hypertrophy [14,15] in response to cytokines and growth factors activated by metabolic factors induced by diabetes [16,17] and obesity [17]. The classic lesion of DKD is nodular glomerular pathology—the nodular Kimmelstiel–Wilson lesions that are observed in about 25% of patients with DKD [11]. These changes are associated with glomerular basement membrane thickening that progresses linearly with the disease duration in T1D. Progression of DKD is influenced by extraglomerular lesions, including tubulointerstitial disease that includes tubular atrophy, interstitial inflammation, and fibrosis. Considered a type of microangiopathy, DKD is also characterized by hyalinosis in the afferent and efferent arterioles [8].

Pathogenic mechanisms leading to DKD are multifactorial and poorly understood, and are triggered by changes in metabolic pathways induced by the diabetic milieu, including an excess in circulating glucose and fatty acid concentrations. The prevailing hypothesis is that alterations in glomerular hemodynamics precede albuminuria and the decline in renal function. Single nephron intraglomerular hypertension leads to hyperfiltration. An elevation in GFR is observed early in 10–67% and 6–73% of patients with T1D and T2D, respectively. Both vasodilation of the afferent arteriole and vasoconstriction of the efferent arteriole contribute to intraglomerular hypertension [18]. In addition, diabetes-associated tubular hypertrophy and increased proximal tubular reabsorption reduce intratubular pressure that favors hyperfiltration by increasing the net filtration pressure [19]. During DKD progression, the glomerular hyperfiltration persists in order to compensate for the progressive reduction in nephron numbers. Increased glomerular filtration in single remnant nephrons is proposed to accelerate the decline in kidney function towards ESRD.

Glomerular basement membrane thickening is an early histopathological finding in DKD, and caused by the decreased turnover of extracellular matrix produced by activated endothelial cells and podocytes. Hyperglycemia stimulates mesangial cell proliferation and matrix collagen synthesis under the control of the transforming growth factor-β (TGF-β) and vascular endothelial growth factor (VEGF) [20]. Collagen degradation is impeded by downregulation of the matrix metalloproteinases [21].

Diabetes induces podocyte foot process effacement and loss, with the latter correlating with albuminuria and the decline in eGFR [22,23]. The podocyte protein, nephrin, involved in cellular signaling and survival [24], is reduced in DKD due to increased nephrin endocytosis [25]. Podocyte loss is compensated for by podocyte hypertrophy, an adaptation that eventually becomes inefficient [26]. Podocyte-specific abnormalities mimic the diabetes-like phenotype with glomerulosclerosis and tubulointerstitial fibrosis in the absence of hyperglycemia [27]. Loss of podocytes is a strong predictor of DKD progression in diabetic patients [28] and murine models of diabetes [23].

Because DKD has been considered a glomerular disease, alterations in the renal tubulointerstitial compartment have been largely overlooked. Tubulointerstitial changes are considered the determining factors of renal function deterioration and progression towards ESRD. Kidney tubular cells represent the largest resident population in the kidney. In DKD, kidney tubule dysregulation may precede or parallel changes of the glomerulus. Only a few days of hyperglycemia leads to tubular hypertrophy. In addition, thickness of the tubular basement membrane is an early structural change observed in normoalbuminuric diabetic patients [29], and is proposed to be an accurate indicator of the severity of DKD [30]. Additional pathologic changes in the tubulointerstitial compartment during the progression of DKD are represented by inflammation, tubular atrophy, peritubular capillary rarefaction, and fibrosis. About 7% of T2D patients present non-functioning atubular glomeruli due to tubular atrophy [31]. Hyperglycemia may directly cause acute tubular necrosis or apoptosis, epithelial-mesenchymal transition, and extracellular matrix deposition. In rodents, the extent of interstitial inflammation correlates with the decline in renal function; inhibition of the interstitial recruitment of inflammatory cells protects rodents from experimental DKD [32]. Genetic or pharmacological inhibition of inflammation improves markers of DKD in diabetes rodent models [33]. In advanced stages of DKD, tubulointerstitial and glomerular changes unite into fibrosis.

In contrast with the DKD with albuminuria, the pathogenesis of the DKD without albuminuria involves macrovascular lesions of intrarenal arteries. This hypothesis is based on the association with T2D [34] and several cardiovascular risk factors in addition to hyperglycemia such as central obesity, hyperlipidemia, and hypertension [35]. The presence of tubulointerstitial lesions including inflammation, tubular atrophy, and an accelerated tubular senescence-like phenotype [36] are reported independent predictors of eGFR decline [37,38].

## 3. Crosstalk between Glomeruli and Tubules in the Diabetic Kidney. Relationship with Kidney Bioenergetics

The crosstalk between glomeruli and tubules is provided by the glomerulo-tubular balance and tubulo-glomerular feedback (Figure 1). The fractional reabsorption of fluid and solutes (sodium and chloride) in the proximal tubules is maintained constant via glomerulo-tubular balance; changes in the glomerular filtration are offset by changes in tubular reabsorption. Glomerulo-tubular balance is independent of neuronal and systemic hormonal control, and mediated by local intraluminal and peritubular factors. An increased glomerular filtration enhances the tubular luminal flow with secondary fluid shear stress that amplifies solute and water proximal tubular reabsorption. In addition, an increase in the filtration fraction enhances the peritubular capillary oncotic pressure that favors proximal reabsorption. As these processes take place via transcellular routes rather than paracellular pathways, they consume energy and involve multiple apical transporters driven by the basal Na-K ATPase.

Tubulo-glomerular feedback regulates glomerular filtration. Normally, increased delivery of sodium, chloride, and potassium to the distal nephron activates the sodium/potassium/chloride transporter in the macula densa cells, causing depolarization and release of adenosine that activates adenosine receptor A1 on the afferent arteriole, causing vasoconstriction and a reduction in glomerular filtration [39,40]. In the context of diabetes, an increase in solute tubular reabsorption upstream of the macula densa (proximal tubule) will increase glomerular filtration.

Proximal sodium reabsorption facilitates glucose reabsorption that involves glucose uptake into the proximal tubular cell via the Sodium-Glucose Transporters 2 and 1 (SGLT2 and SGLT1, respectively) followed by its exit from the cell via GLUT2 on the basolateral membrane. The high-capacity transporter, SGLT2, accounts for about 97% of renal glucose reabsorption, whereas the low-capacity SGLT1 reabsorbs the remaining 2–3% of glucose in the late proximal tubule [41,42]. SGLT2 uses one sodium ion while SGLT1 uses two sodium ions to transport one glucose molecule, indicating that SGLT2 are more energy efficient than SGLT1. Sodium reabsorption by SGLT1 and SGLT2 is electrogenic and is thought to drive chloride reabsorption by electrodiffusion. Sodium, chloride, and glucose reabsorption drive water reabsorption. Mathematical modeling predicted that in diabetes, the glomerular filtration of the excess of glucose and other solutes, including sodium, is followed by an increase in their proximal tubular reabsorption to maintain the glomerulo-tubular balance, which will reduce the solute concentration in the macula densa, thus increasing the glomerular filtration rate through tubulo-glomerular feedback [43].

The effect of hyperglycemia to increase the glucose glomerular filtration and proximal tubular reabsorption was confirmed in T1D and T2D patients [18]. The diabetic-induced increase in proximal glucose and sodium reabsorption is facilitated by multiple factors including tubular hypertrophy [18], upregulation of the SGLT2 [44,45] and GLUT2 in both diabetic animals [46,47] and human patients [48,49], and their membrane recruitment from cytosolic pools [50]. In advanced DKD, glomerular hyperfiltration and tubular reabsorption are amplified by the increased work load of the remaining intact nephrons to compensate for the nephron loss [51]. Insulin-dependent SGLT2 phosphorylation enhances its transport activity [52] and may count for the increased glucose tubular reabsorption in T2D. Finally, high glucose upregulates the basolateral GLUT1 that facilitates glucose uptake and metabolism [53,54].

Glomerular hyperfiltration imposes the highest transcellular transport burden and energy consumption by the proximal tubules, indicating an increase in ATP demand and oxidative metabolism in the renal cortex. Na-K ATPase, the biggest ATP consumer, located on the basolateral membrane of tubular cells, decreases intracellular sodium concentration and therefore provides the driving force for sodium uptake across the apical membrane. Na-K ATPase also drives the apical sodium-hydrogen exchanger that is considered a determinant of tubular reabsorption, kidney weight, and GFR [55]. Activation of the Na-K ATPase reduced diabetic glomerular hyperfiltration [56] and decreased kidney hypertrophy [57].

Mathematical modelling predicts that sodium transport increases by 50% while sodium transport-dependent oxygen consumption increases by 100% in the diabetic rat kidney, with proximal tubules being the highest consumers [58]. In addition, diabetes increases oxygen consumption in the kidney outer medulla [58,59,60]. In agreement with these findings, oxygen tension in the renal cortex was reduced in a rat model of T1D [60]. Inhibition of SGLT2 in the diabetic kidney reduces oxygen consumption in the proximal tubule and renal cortex due to lowering GFR [58,59].

In conclusion, the energy consumed for glomerular filtration and tubular reabsorption is higher in diabetes compared to normal subjects. Preservation of renal oxygenation and integrity of energy-producing structures are critical for maintaining kidney function in diabetes [61].

## 4. Mitochondrial Energy Metabolism in the Normal Kidney Cells

The kidney has one of the highest resting metabolic rates in the body, and secondary to the heart, the highest mitochondrial content and oxygen consumption rates. The increased mitochondrial density supports high energy production in the form of ATP, which must meet the high energy requirement needed for urine formation. In addition to the canonical role of ATP provider, mitochondria are essential in regulating oxidative stress, calcium cellular metabolism, redox balance, and cell death [62].

The major source of kidney ATP is mitochondrial oxidative phosphorylation (Figure 2) with a small amount deriving from glycolysis. The different segments within the nephron have specific metabolic profiles. The cortex, with a large number of proximal tubules, is the main site of fatty acid (FA) oxidation and renal gluconeogenesis, while the medulla has higher rates of glycolysis [63]. Proximal and distal tubules, located within the cortex, are rich in mitochondria, whereas tubular cells of the collecting duct and loop of Henle, located within the medulla, have fewer mitochondria [64,65]. Podocytes have intermediate mitochondrial density and exhibit a flexible phenotype using both oxidative phosphorylation and glycolysis for energy needs [66,67,68].

As proximal tubules use active transport for reabsorption, their mitochondria must be able to detect and respond to fluctuations in nutrient availability and energy demand [62]. They obtain ATP by oxidative processes using a plethora of substrates (FA, lactate, citrate) other than glucose [72]. Mitochondrial FA β-oxidation is the major energetic source in the proximal tubules [63], as this pathway is more efficient by generating a larger amount of ATP. For example, one molecule of palmitate generates 106 molecules of ATP while the oxidation of glucose produces 36 ATP molecules. Although glomeruli also rely on ATP for house-keeping cell functions, glomerular filtration is less energy consuming as it uses the net hydrostatic pressure instead of ATP to support glomerular filtration. Therefore, glomerular cells—podocytes, mesangial, and endothelial cells—depend on glucose as the major respiratory fuel [62]

Glucose enters renal cells via glucose transporters (GLUTs) (Figure 2) and is directed through multiple metabolic pathways such as glycolysis, glycogen synthesis, polyol, hexosamine biosynthetic, or pentose phosphate pathways. The end product of glycolysis, pyruvate, is either converted to lactate or transported into mitochondria via the mitochondrial pyruvate carrier, and converted by pyruvate dehydrogenase (PDH) to acetyl-CoA for the tricarboxylic acid (TCA) cycle.

Long chain FAs (i.e., palmitate) are activated to FA-Coenzyme A products and enter the mitochondria via carnitine palmitoyltransferases (CPT1 and 2) to be oxidized via FA β-oxidation. The end products of each FA β-oxidation cycle are NADH, FADH_2_, and acetyl-CoA, which are further oxidized by the electron transport chain (ETC) complexes or TCA cycle, respectively, ultimately leading to ATP synthesis via mitochondrial oxidative phosphorylation. While electrons are transferred from the reducing equivalents, NADH and FADH2, to oxygen by the ETC complexes, an electrochemical gradient is built across the mitochondrial inner membrane (IM), which is used by the ATP synthase (complex V) to phosphorylate ADP and form ATP. Mitochondrial ATP is transferred to the cytosol by phosphate exchange networks including mitochondrial and cytosolic creatine kinases to support different cellular functions (Figure 2).

## 5. Mitochondrial Abnormalities in the Diabetic Kidney

Mitochondrial alterations precede the development of albuminuria and renal histological changes in diabetes. For example, mitochondrial fragmentation and a decreased ATP content occurred in proximal tubular cells in early (4 weeks) diabetes in the absence of albuminuria and specific glomerular pathology. Eight weeks of diabetes progressed towards mitochondrial permeability transition pore opening associated with mitochondrial-generated oxidative stress and glomerular alterations. Sixteen weeks of diabetes progressed towards evident tubular injury associated with an increase in complex I-dependent oxidative phosphorylation and decreased ATP content, indicating mitochondrial uncoupling. These data support the concept that mitochondrial dysfunction is a pathogenic mechanism in the development of DKD, which progresses in parallel with kidney alterations [73]. In this review, we summarize data showing that diabetes affects all aspects of bioenergetic metabolism including switching the proportions of cell-specific fuel sources, causing enzymatic defects within the electron transport chain (ETC) complexes with consequences on ATP production and redox balance.

### 5.1. Alterations in Substrate Selection

The diabetic milieu provides an excess of energetic substrates in the form of glucose, free fatty acids (FA), amino acids, and ketone bodies.

#### 5.1.1. Glomerular Cells

Glomeruli contain endothelial cells, mesangial cells, and podocytes, all significant for the development of the glomerular diabetic phenotype. Glucose is taken up by glomerular cells via facilitative glucose transporters, GLUTs, including the constitutive GLUT1 and insulin-sensitive GLUT4 (Figure 2). Glomerular endothelial cells have a full insulin signaling pathway and become insulin insensitive when cultured in high glucose and during experimental T1D and T2D [74]. Despite insulin insensitivity, glucose uptake during hyperglycemic conditions increases in endothelial cells via the constitutive GLUT1. Podocytes also have multiple GLUTs, including GLUT1 and GLUT4 [27], and an insulin signaling cascade that is critical to maintain podocyte integrity in models of T2D [75]. In podocytes, GLUT1 regulates the glucose uptake in both basal and insulin-stimulated conditions [27]. Diabetes-induced GLUT1 overexpression caused increased glucose uptake in podocytes and mesangial cells, however, it was proven protective against mesangial expansion and albuminuria only when overexpressed in podocytes, suggesting a crosstalk between glomerular cells [76]. In addition to hyperglycemia, the mechanical stress induced by glomerular hypertension has an additive effect to increase basal insulin-independent glucose uptake [77,78]. GLUT4 mRNA was downregulated in podocytes of T1D diabetic patients with albuminuria compared to those without albuminuria, and considered adaptive [79]. High glucose upregulated GLUT1 and GLUT3 in cultured human podocytes, and increased GLUT1 translocation on the plasma membrane [78]. Protecting podocytes from increased circulating glucose level by inducing a podocyte-specific deletion of GLUT4 prevented the diabetes-induced albuminuria [80]. These studies suggest that the GLUT-mediated glucose uptake is increased and harmful in glomerular cells during diabetes.

Within the cytosol, glucose-6-phosphate is converted to glyceraldehyde 3-phosphate and then to pyruvate via the enzyme glyceraldehyde-3-phosphate dehydrogenase (GAPDH). Because GAPDH was reported as inhibited in endothelial cells exposed to high glucose [81], and mitochondrial oxidative phosphorylation is depressed in diabetes (as described later), glycolytic intermediates follow alternative cytosolic non-ATP-producing pathways such as polyol pathway, activation of protein kinase C, and formation of advanced glycation end products and hexosamine pathway (Figure 2), which are recognized as central to diabetic chronic microvascular complications [81]. During the process of differentiation in a normal glucose environment, human podocytes activate oxidative metabolism and reduce glycolytic enzymes [82], while hyperglycemic conditions promote metabolic reprogramming with a reduction of mitochondrial metabolism and increased glycolysis [82]. A similar glycolytic switch is observed in kidney sections of human patients with DKD. This is consistent with studies showing that mitochondrial function is depressed in patients with DKD [83]. A comprehensive approach with transcriptomics, metabolomics, and metabolic flux analysis in both 12- and 24-week-old db/db T2D mice outlined an increase in glycolysis in the diabetic kidney cortex [84].

Glucose conversion to sorbitol (via aldose reductase with NADPH oxidation) and then to fructose (via sorbitol dehydrogenase with NAD reduction), also the “polyol pathway”, has a very low flux in basal conditions while it increases during hyperglycemia, leading to the relative depletion of cytosolic NADPH and reduced glutathione (GSH) with secondary osmotic and oxidative stress. Increased cytosolic glucose increases diacylglycerol, causing chronic activation of protein kinase C isoforms, from which PKC-β2 has special relevance in the diabetic kidney, as it activates downstream pathways leading to inflammation and extracellular matrix accumulation [85,86]. Excessive glucose and glucose-derived dicarbonyls (i.e., methylglyoxal) react with lysine and arginine amino groups of proteins to form advanced glycation end-products (AGEs). Intracellular targets of methylglyoxal have been reported including mitochondrial proteins [87], indicating a direct effect on energetic metabolism [88]. Intracellular AGEs have a very broad spectra of actions causing cell injury, increased release of cytokines and profibrotic factors, leading to glomerular sclerosis and tubulointerstitial fibrosis. AGEs bind the cell-surface AGE-binding receptors, leading to the modulation of cellular functions, and the activation of pro-oxidant, pro-inflammatory events. The hexosamine pathway provides the end product, UDP-N-acetyl-glucosamine (UDP-GlcNAc), which is a precursor of extracellular matrix proteins such as the proteoglycans, and is used by the enzyme O-GlcNAc transferase to modify proteins by O-GlcNAcylation, thus causing epigenetic modifications and changes in gene expression. An excellent review on the role of hyperglycemia and increased cytosolic glucose metabolism in DKD is provided by Thomas et al. [12].

In conclusion, while glucose uptake is increased in glomerular cells in diabetes, glucose oxidation is decreased and alternative glycolytic pathways in the cytosol are activated, thus triggering pathogenic mechanisms that are involved in diabetic microangiopathic complications.

#### 5.1.2. Tubular Cells

Tubular cells account for approximately 90% of the renal cortex. Proximal tubular reabsorption requires an enormous amount of energy, which is mostly provided by mitochondrial FA β-oxidation and to a lesser extent by glucose metabolism. The proximal tubule is also the second most important organ after the liver for gluconeogenesis [62].

Diabetes [89] and high-glucose conditions upregulate proximal tubule SGLT2 and GLUT2 [48,90], and activate the Na/K ATPase activity [89], leading to an increased cortical oxygen consumption and reduction in the cortical oxygen partial pressure, which were reversed by SGLT2 inhibition [60]. These data indicate that proximal sodium and glucose reabsorption require enhanced oxidative metabolism. A decrease in the cortical oxygen pressure upregulates the hypoxia-inducing factor (HIF1) α, which also enhances GLUT1 mRNA expression that favors glucose uptake in the tubular cells.

Proximal tubules reabsorb the glomerular-filtered glucose and can synthesize glucose. Under normal glucose conditions, the reabsorbed glucose is not used for ATP production as it is readily reabsorbed on the basolateral side back into the blood. Different pathological conditions increase the glycolytic metabolic flux and expression of glycolytic enzymes in the kidney proximal tubules [79,87,88]. A metabolic shift from oxidative phosphorylation to glycolysis has been proposed to occur in damaged renal tubules such as regenerating proximal tubule cells after acute kidney injury, proximal tubular cells undergoing atrophy [91], and tubules in DKD [71].

In the human diabetic kidney, the fate of the urine-derived glucose has yet to be determined. Both SGLT2 and GLUT2 are increased in diabetic tubules, indicating increased glucose transport rates to prevent cellular glucose accumulation; however, the exact ratio of the SGLT2-mediated glucose uptake to GLUT2-mediated glucose export in human tubules has yet to be determined. It is suggested that if the glucose concentration is higher on the basolateral side, then the intracellular glucose levels in proximal tubular cells must be higher to establish the necessary GLUT2 gradient.

However, glycolysis is not the major metabolic pathway in proximal tubules. It is reported that here glucose-derived fructose from the polyol pathway is further converted by the enzyme ketohexokinase to fructose-1-phosphate [92,93], which is incorporated within metabolism. The deficiency of ketohexokinase protected against tubular damage in diabetic mice, indicating a role in fructose metabolism in the pathogenesis of DKD [94]. The intensified polyol pathway is reported to lead to ATP depletion in proximal tubules upon high glucose conditions [62].

Similar to cardiomyocytes [95], renal epithelial cells depend on mitochondrial FA β-oxidation as energy sources [96] rather than glucose oxidation [97]. However, research into lipid metabolism in the diabetic kidney are limited. Normally, FA uptake, oxidation, and synthesis are tightly balanced to avoid intracellular lipid accumulation. Clinical observations indicate an association between diabetic hyperlipidemia and DKD development [98], suggesting that lipid accumulation in the tubules (lipotoxicity) is detrimental.

Most of the studies conclude that an increased FA β-oxidation, which matches the increased circulating free FA and increased FA uptake, is critical for tubular cell integrity. The major route for FA uptake in kidney cells is provided by the multifunctional transmembrane glycoprotein CD36, which is highly expressed in all kidney cells [99], dependent on peroxisome proliferator-activated receptor (PPARγ) [100], and facilitates the pro-apoptotic effect of palmitate on podocytes [101] and renal tubular cells [102]. CD36-mediated FA uptake is increased in DKD experimental models [103]. Renal tubular epithelial cells cultured in high-glucose conditions exhibited an increased CD36 expression, which was associated with triglyceride accumulation [104]. In addition, diabetic kidney presents an increased LDL-mediated cholesterol uptake associated with downregulation of genes effecting cholesterol efflux, leading to increased intracellular lipid accumulation.

We observed an increased FA β-oxidation in diabetic tubular mitochondria from T1D rats [105]. In contrast, in human subjects with more advanced diabetes, genes involved in mitochondrial FA β-oxidation were found downregulated [106], which was significantly correlated with reduced eGFR and interstitial inflammation, and suggest that FA β-oxidation may follow a bi-modal pattern. Overexpression of enzymes involved in FA synthesis [107] associated with a decrease in FA β-oxidation enzymes [108] may contribute to FA accumulation in diabetic tubules. This is supported by the observation that preventing lipid accumulation by increasing lipid utilization delayed DKD progression [96]. A decreased FA β-oxidation with secondary ATP depletion in renal tubular cells caused cell death and reprogrammed the cells into dedifferentiation towards a profibrotic phenotype [96].

In kidney fibrosis there is a decrease in FA oxidation in tubular cells without an equal increase in glucose oxidation, so there is no real switch in the energy substrate preference. Because there is no energetic switch, the energy deficit is quite logical and is considered a major pathogenic mechanism for kidney fibrosis in both humans and animal models [96]. While tubular cells may not be the direct precursors of interstitial fibroblasts, they play a critical role in fibrosis by multiple mechanisms including secreting cytokines [109]. The profibrotic cytokine, TGF-β1, long time considered downstream in the fibrosis process, appears to be an upstream modulator of FA metabolism in renal tubular epithelial cells. TGFB1-driven Smad3-dependent pathway inhibits mitochondrial metabolism and fat accumulation [110].

The decrease in FA oxidation may be part of a broader metabolic reprogramming in tubular cells or a reflection of mitochondrial electron transport chain (ETC) defects, which will be addressed later in this review. In support of the concept of a broader mitochondrial dysfunction, in addition to lipid oxidation, carbohydrate and amino acid metabolisms were also depressed in patient samples with kidney fibrosis [96].

In conclusion, FA uptake is increased in tubular cells in diabetes while their mitochondrial β-oxidation follows a bi-modal pattern with an initial increase and a later depression, with the latter not being matched by an equal increase in other energy-generating pathways, potentially leading to energy starvation.

### 5.2. Mitochondrial Biogenesis

Producing new functional mitochondria via mitochondrial biogenesis provides a reserve to increase ATP production with the purpose to meet increased energy demands. The peroxisome proliferator-activated receptors (PPARs) and PPARγ coactivator-1α (PGC1α) are key transcription factors that regulate the expression of proteins involved in FA uptake and oxidation in the kidney [111], TCA cycle, and oxidative phosphorylation [112] (Figure 3). Whereas upregulation of PGC1α has been observed in the early phase of diabetes [113], PGC1α was suppressed in long-term diabetes and kidney tissues of patients with DKD [83,114,115,116] and other pro-fibrotic diseases [117]. The concept that PGC-1α has a critical role in DKD is supported by the report that PGC-1α overexpression in renal tubular cells improved kidney pathology in mouse models of DKD by restoring the defective FA oxidation and correcting ATP depletion [96].

PGC1α expression is regulated by external and intracellular stimuli as well as posttranslational modifications, and is considered to be a nutrient sensor in the kidney. For example, the taurine upregulated gene 1 (Tug1), a long noncoding RNA, increases PGC1a expression, mitochondrial complex I, oxidative phosphorylation, and ATP generation. Tug1 is decreased in podocytes of diabetic mice and glomeruli of patients with DKD, and its overexpression protected mice against DKD [116]. PGC1α is inactivated by acetylation of its lysine residues. SIRT1-mediated deacetylation of PGC-1α mitigated podocyte injury while resveratrol, an activator of SIRT1, protected mitochondrial function against podocyte injury [118]. In turn, PGC-1α upregulates mitochondrial SIRT3, which has also been implicated in progression of DKD [119,120]. These data show that PGC-1α, the master regulator of mitochondrial biogenesis, may be an important crossing point of pathways that maintain mitochondrial integrity and a potential target for therapy in DKD.

### 5.3. Mitochondrial ETC Defects

A strong association between mitochondrial defects and kidney dysfunction is provided by the observation that human congenital nephrotic syndrome is associated with decreased mitochondrial transcription for ETC subunits leading to ETC functional defects and oxidative stress in kidney glomeruli [121]. Kidneys of human subjects with DKD show markers of oxidative damage of mtDNA [122], which encodes for several ETC protein subunits. Studies of urine metabolomics show decreased mitochondrial-derived metabolites and mtDNA content within urinary exosomes, associated with decreased mitochondrial biogenesis markers and proteins in human diabetic kidneys [83]. Metabolomic studies identify the TAC cycle, pyrimidine metabolism, amino acids, propionate, fatty acid, and oxalate metabolism as metabolic pathways affected by diabetes [123]. These observations in diabetic human subjects suggest that mitochondrial dysfunction and the inability to meet the increased energy demand may be a key pathogenic factor in the progression of DKD [124].

Mitochondrial ETC defects are consistent findings in DKD. Previous reports linking progression of DKD with ETC defects identified depressed complex I, III, and/or IV activities in mitochondria from either whole diabetic kidney or cortex in animal models of DKD [84,115,125,126,127,128]. Whereas complex I activity seemed to be increased in the early phase of diabetes in some studies [125,129], a large body of evidence has shown that complex I, III, and IV activities are reduced as DKD progresses [115,126,127,128,130]. The reduced activity of complex I, the largest complex in the ETC, is the most frequent finding in diabetic glomeruli and podocytes [130,131]. In kidneys of diabetic human subjects, complex IV staining score was significantly lower compared with those of non-diabetic patients [83], and the protein level of the complex IV subunit, mitochondrially encoded cytochrome c oxidase II, was decreased in glomeruli isolated from T2D patients with lower eGFR compared to those isolated from patients with normal eGFR or nondiabetic subjects [132]. These data causally link the ETC defects with the decrease in eGFR in DKD.

Oxidative phosphorylation assessed as oxygen consumption rates (OCR) shows a bimodal pattern over the course of diabetes. Oxygen consumption rates were increased during very early phases of experimental diabetes in the renal cortex and proximal tubular cell mitochondria [133,134,135,136], followed by a decline with the progression of albuminuria. We also reported an increased OCR with FA substrates in tubular mitochondria [105]. In contrast, OCR in glomeruli and podocytes were decreased, disregarding the phase of diabetes [112,118,131]. It is suggested that the increase in OCR during the early phases of DKD may reflect a metabolic reprograming in response to the excess of metabolic substrates in diabetes [137]. This pattern that is observed in tubules rather than glomeruli may be an adaptation that resembles the bioenergetic changes of cardiomyocytes exposed to excessive energetic substrates, i.e., metabolic syndrome.

In glomeruli, a bioenergetic crosstalk between cells is observed to be critical for the progression of DKD. Podocyte-restricted injury led to mitochondrial dysfunction and oxidative stress in glomerular endothelial cells. The relationship is mutual. Mitochondrial dysfunction in glomerular endothelial cells cause endothelial damage, albuminuria, podocyte depletion, and glomerulosclerosis [122].

The impact of mitochondrial dysfunction is determined by the metabolic heterogeneity in different segments of the nephron with proximal tubules generating ATP mainly via mitochondrial oxidative phosphorylation, whereas podocytes, endothelial, and mesangial cells relying on glycolysis [138]. The consequences of the ETC defects on the bioenergetic state are inconsistent. Coughlan et al. reported a decreased ATP content in proximal tubules of the diabetic rats [73], whereas other studies observed that ATP content was either unchanged, concordant with no change in oxidative phosphorylation [73], or even increased [139], in agreement with the increase in mitochondrial proteins involved in energetic substrate oxidation. The other potential consequences of mitochondrial defects, including oxidative stress and dysregulation of the redox balance (Figure 3), will be discussed in further sections.

### 5.4. Mitochondrial Oxidative Stress

Reactive oxygen species (ROS) are end-products of aerobic metabolism. Superoxide (O_2_), generated by the one-electron reduction of oxygen, can either oxidize neighboring molecules or be dismutated to hydrogen peroxide (H_2_O_2_) by superoxide dismutase enzymes, SOD. The hydroxyl radical, generated via the Fenton reaction from superoxide and hydrogen peroxide, can initiate the formation of lipid and lipid peroxyl radicals. H_2_O_2_ is removed by antioxidant enzymes including peroxiredoxins, catalase, and glutathione peroxidases. An increase in ROS generation compared to their neutralization by antioxidant mechanisms is defined as oxidative stress [140], which is considered a major pathogenic mechanism for chronic diabetic complications including DKD. Strong evidence for the involvement of oxidative stress in DKD is provided by the finding that oxidative stress is decreased in the kidney cortex of DKD-resistant diabetic mice [115].

The ROS-generating sources in the kidney are both extramitochondrial and mitochondrial. Eleven sites within the mitochondrial ETC and redox enzymes are able to leak electrons and cause one- or two-electron reduction of oxygen to form O_2_ and H_2_O_2_ [141]. These redox centers become sites of electron leakage when they are induced in a maximally reduced state due to either impairment of electron flux or redistribution of electrons as a result of defects in downstream catalytic subunits.

Different energetic substrates engage different dehydrogenases to generate reducing equivalents that further use different entry points within the ETC. Therefore, it is expected that glomerular and tubular mitochondria have different sites of ROS generation as they use different energetic substrates in physiological conditions and experience changes in substrate preference in diabetes. In addition, diabetes causes ETC defects that impede the electron flux, thus causing electron accumulation at various redox centers.

Pyruvate, the end product of glycolysis—a major bioenergetic pathway in glomerular cells—is converted to acetylCoA by pyruvate dehydrogenase, that becomes an important site of electron leak upon slow electron flux [142]. The long chain FA-CoA, palmitoyl-CoA, the major energetic substrate for proximal tubules, provides electrons at three ETC entry sites, complex I (via the reduction of NAD by hydroxyacyl-CoA dehydrogenase and the dehydrogenases of the TAC cycle from acetylCoA), electron-transferring flavoprotein (ETF) ubiquinone oxidoreductase via its reduction by ETF, and to a minor extent complex II (via the formation of succinate in the TAC cycle). In normal mitochondria and absence of ADP to consume the electrochemical inner membrane gradient, sites in complex I, II, and III are the main contributors to mitochondrial O_2_/H_2_O_2_ production, with ETF pathway being negligible. The contribution of complex II increases in the presence of electron backflow from the ubiquinone pool into complex II rather than by forward flow from succinate generated by the TCA cycle. In contrast, in diabetes, tubular mitochondrial FA β-oxidation becomes the major source of ROS generation from the ETF-ETF-QR pathway [105].

Caloric restriction is beneficial on cellular redox homeostasis, an effect that is mediated by an increase in the reduced glutathione (GSH)—a major antioxidant, rather than by a decrease in the mitochondrial ROS generation [143]. In addition, low glucose may lead to an increase in the oxidized mitochondrial NAD pool and a concomitant decrease in mitochondrial protonmotive force [144,145], which both decrease the mitochondrial ROS production from sites in complex I [146,147] and upstream dehydrogenases [148]. In contrast, overnutrition and high glucose conditions are suggested to increase the reduced NAD (NADH) [145,149] with higher mitochondrial ROS generation. These data suggest that an excess of energetic substrates increase NADH and the protonmotive force [149] that both enhance ROS generation by decreasing the electron flow within the ETC.

Mitochondrial uncoupling protein 2 (UCP2), which transports protons across the inner membrane to the matrix, is expressed in the kidney proximal tubules [150], activated by mitochondrial ROS to dissipate the proton motive force and reduce the O_2_ production while protecting glomerular damage in experimental diabetes [133,151]. Studies investigating the association between UCP2 polymorphisms and increased risk to develop DKD in human subjects suggest that UCP2 may be a potential target for treatment [152].

The consequences of oxidative stress on kidney are manifold and include enhancing profibrotic and inflammatory signals, such as protein kinase C, mitogen-activated protein kinases, cytokines, and transcription factors. Mitochondrial-generated ROS oxidize mitochondrial proteins and mtDNA, thus entertaining a vicious cycle of mitochondrial damage and dysfunction (Figure 3).

### 5.5. NADH/NAD Redox State

#### 5.5.1. NAD Pool

Nicotinamide adenine dinucleotide (NAD) shuttles electrons from glycolysis and the TCA cycle to complex I in the ETC. The oxidized form, NAD^+^, is also a co-substrate for non-redox reactions such as those catalyzed by NAD-dependent lysine deacetylases sirtuins (SIRTs); poly (ADP-ribose) polymerase (PARP); and the NADase, CD38 [153,154]. Maintaining the NAD pool is critical for sustaining mitochondrial function and preventing kidney damage [155], and protecting against obesity and metabolic syndrome [156]. However, the upregulation of the nicotinamide phosphoribosyltransferase, a key enzyme in the NAD biosynthetic pathway, is reported to promote kidney fibrosis [157], suggesting that the ratio between the reduced and oxidized forms (NADH/NAD^+^ ratio) is more significant for the kidney metabolism than the total NAD pool.

#### 5.5.2. NADH/NAD^+^ Redox Ratio

Cellular metabolism is based on redox reactions that involve electron transfer between reduced and oxidized compounds. A common example is the transfer of electrons in the mitochondrial ETC from reduced to oxidized subunits. There are four major redox couples (redox players) [158] that reflect the cellular redox status, and are involved in redox signaling: NAD^+^ (oxidized)/NADH (reduced), NADP^+^/NADPH, GSSG (glutathione disulfide)/GSH (glutathione), as well as TrxSS (oxidized disulfide thioredoxin)/TrxSH2 (reduced thioredoxin). The increased flux through the polyol pathway (“hyperglycemic pseudohypoxia” hypothesis [159,160]) in diabetes utilizes NADPH as an electron donor and NAD^+^ as an acceptor to become NADH, thus increasing the NADH/NAD^+^ ratio and depleting the antioxidant cofactor NADPH [161].

Another potential mechanism that may increase cellular NADH/NAD^+^ redox ratio is the activation of poly (ADP-ribose) polymerases (PARPs) involved in DNA repair. When activated in response to DNA damage, PARPs consume NAD^+^, thus causing NAD^+^ depletion and altering redox homeostasis. PARP-deficient mice were protected against diabetes [162] and preserved redox homeostasis and mitochondrial function [163] by activating the NAD^+^-dependent deacetylases, sirtuins (SIRTs).

Cellular organelles are characterized by specific redox states. For example, cytosolic NADP^+^/NADPH is maintained in a more reduced state that is necessary to drive antioxidant mechanisms. In energized mitochondria, NADH exceeds NAD^+^ while the cytosol has a higher NAD^+^ [95]. Mitochondrial NADH/NAD^+^ redox ratio increases in mitochondrial dysfunction [164]. Complex I and IV defects caused NADH accumulation [165,166], and corrections of ETC defects normalized NADH and restored redox balance [167]. Because glucose restriction increased the mitochondrial NAD^+^ with a concomitant decrease in mitochondrial protonmotive force [144,145], it is hypothesized that an excess of energetic substrates, glucose, and FAs, generate excessive NADH by mitochondrial oxidation. These data suggest that diabetes creates conditions that favor a reduced redox microenvironment in mitochondria, which is detrimental to kidney function. Kidneys of T2D rats showed a reduced NAD^+^/NADH ratio. CD38 inhibition increased NAD and reduced tubulointerstitial fibrosis, tubular cell damage, and inflammatory gene expression in diabetic rats [168].

The immediate consequence of an increased mitochondrial NADH/NAD^+^ redox ratio is a reductive stress (increased NADH) and a deficiency in the oxidized form, NAD^+^, leading to impairment of enzymatic activities that use NAD^+^ as a co-factor. Among the large family of NAD^+^-dependent deacetylases, SIRTs, SIRT1 is an extramitochondrial protein that is expressed in tubular cells with a wide range of functions in metabolism and sodium reabsorption [169]. SIRT1 expression is decreased in kidneys of human patients with DKD [170] and is an early event occurring before albuminuria in proximal tubules of T1D and T2D diabetic mice [171]. SIRT1 upregulation in proximal tubules [171] and podocytes [172] decreased diabetic kidney injury, while SIRT1 knockout led to albuminuria in non-diabetic mice and exacerbated diabetic changes in the glomeruli of both streptozotocin and db/db diabetic mouse models [171]. The mitochondrial SIRT3 maintains mitochondrial energy homeostasis and antioxidant defense in kidney tubule [173] and is protective against DKD [174,175]. Decreased NAD^+^/NADH ratio and SIRT3 activity in mesangial cells exposed to high glucose led to oxidative stress and mesangial hypertrophy [176] while in tubules of diabetic rats they caused lysine acetylation and inactivation of the Mn-SOD causing oxidative stress [177]. SIRT3 overexpression ameliorated high glucose-induced oxidative stress and apoptosis in cultured human tubular cells [178]. In DKD with fibrosis, SIRT3 is suppressed and associated with increased glycolysis, ROS generation, inflammation, and fibrosis. SIRT3 deficiency coupled with suppression of PGC1α [179] while PGC1a deficiency suppressed SIRT3 and caused mitochondrial defects [180] (Figure 3). Inhibition of glycolysis rescued both PGC1α and SIRT3 in the diabetic kidneys, suggesting that increased glycolysis impairs the PGC1α –SIRT3 axis to protect mitochondrial metabolism. SIRT6 is reported to maintain podocyte integrity and function, and the impermeability of the glomerular filtration membrane to proteins [181]. The FA β-oxidation enzymes are regulated by reversible posttranslational modifications, including lysine acylation. The mitochondrial SIRT5 reversed lysine acylation modifications on several FA β-oxidation enzymes. SIRT5 knockout favors peroxisomal FA oxidation and protected tubules in a model of acute tubular injury.

While it is accepted that the redox imbalance drives the transition from reductive stress to oxidative stress [182], the direct measure of NAD and NADH in different kidney cells during the progression of diabetes has not been reported. As DKD progression is assessed by albuminuria and eGFR, a correlation between redox stresses and these DKD progression criteria has not been investigated. The oversupply of energy nutrients has the potential to increase the NADH/NAD^+^ redox ratio in parallel with the severity of DKD. More studies are needed to quantify this ratio and dissect the role of each player in this cascade of biochemical redox reactions. Therefore, future efforts should be made to assess redox imbalance-induced damage to different kidney cells in diabetes.

The mitochondrial NADH/NAD^+^ and NADPH/NADP^+^ redox couples are linked by nicotinamide nucleotide transhydrogenase (NNT) (Figure 4), an enzyme that leverages the proton-motive force in the oxidation of NADH and simultaneously reduces NADP^+^. NNT maintains a NADPH/NADP^+^ ratio several-fold higher than the NADH/NAD^+^ ratio, and thus is a physiologically relevant source of NADPH that drives the reduction of H_2_O_2_ [183]. While NNT is reported to be expressed exclusively in cardiac tissue [184], other studies report it in the kidney [185]. The role of NNT to maintain the mitochondrial redox state and antioxidant defense in the diabetic kidney cells is yet to be determined. As NNT has the potential to decrease NADH and recover NAD, future research is needed to determine its role in normalizing the NADH/NAD^+^ redox state in DKD.

### 5.6. Mitochondrial Dynamics and Quality Control

Mitochondria are dynamic organelles that must undergo remodeling via fission and fusion to meet the cellular energy demand. [188]. Specific proteins regulate mitochondrial dynamics, with members of the dynamin superfamily GTPases Drp1 and mitochondrial fission protein 1 controlling fission, and mitofusion 1, mitofusin 2 (MFN1 and 2, respectively), and optic atrophy 1 (OPA1) proteins regulating fusion [189]. Fusion with elongated mitochondria is associated with increased oxidative phosphorylation and more efficient exchange of material between healthy mitochondria, while the splitting of mitochondria during fission allows for the segregation and elimination of damaged organelles [62]. Errors in fission and fusion as well as an imbalance between the two processes lead to mitochondrial dysfunction. For example, deletion of MFN2 in mice results in coenzyme Q deficiency leading to reduced ATP production [190]. In addition, excessive fission alters mitochondrial homeostasis and is reportedly involved in the pathogenesis of diabetic nephropathy [62]. High glucose increases the expression of the Rho-associated coiled coil containing protein kinase 1 (ROCK1) activity in podocytes, leading to Drp1 phosphorylation, which promotes mitochondrial fission and enhances ROS generation. In contrast, genetic deletions of Drp1 in diabetic T2D db/db mice decreased fission and improved mitochondrial fitness in podocytes, and protected against progression of DKD by mitigating albuminuria, mesangial matrix expansion, glomerular basement membrane thickness, and podocyte effacement. Mitochondrial fission under hyperglycemic conditions is associated with changes in mitochondrial morphology with small and round mitochondria, mitochondrial debris and impaired mitochondrial DNA distribution, decreased expression of ETC complex subunits and ATP generation, opening of the mitochondrial permeability pore, and mitochondrial-induced apoptosis [191]. As an important component of the diabetic milieu, the excess of fatty acids (FA) changes mitochondrial dynamics. Cultured mouse podocytes cultured with an excess of palmitic acid show an increase in Drp1 and mitochondrial fragmentation leading to podocyte injury [192]. Similarly, human renal proximal tubular cells treated with high glucose show a downregulation of MFN1 and an increase in DRP1, both promoting mitochondrial fission [193].

A regulator of multiple cellular functions, the dual specificity protein phosphatase-1 (DUSP1) is a threonine-tyrosine phosphatase that dephosphorylates and inactivates extracellular stress-regulated kinase (p38) and c-Jun N-terminal kinase (JNK), with the latter favoring mitochondrial fission. Chronic hyperglycemia downregulates DUSP1, and is associated with kidney hypertrophy, renal fibrosis, and glomerular apoptosis. Defective DUSP1 activates the JNK pathway that leads to mitochondrial fission.

Chronic hyperglycemia upregulates the nuclear receptor subfamily 4 group A member 1 (NR4A1) that activates p53 to stimulate mitochondrial fission and represses Parkin protein to inhibit mitophagy. In this context, the defective Parkin-mediated mitophagy represses ATP production while the cell is unable to limit mitochondrial fission. Alterations in Parkin-mediated mitophagy is a key feature in DKD [194].

Mitophagy is a physiological process by which damaged mitochondria are cleared by being engulfed by autophagosomes, and subsequently degraded by lysosomes [195]. The process is either ubiquitin-dependent or -independent. The ubiquitin-dependent pathway is regulated by the phosphatase and tensin homolog-induced putative kinase 1 (PINK1) and Parkin [195,196]. PINK1 is normally transported into mitochondria and degraded. However, once mitochondria are damaged and depolarized, PINK1 and Parkin accumulate on the outer membrane, leading to the autophagic degradation of mitochondria. PINK1 recruits and phosphorylates both the E3-ligase Parkin and ubiquitin to create the poly-ubiquitin chains to serve as the recognition signal for autophagic proteins and mitochondrial destruction. Within the ubiquitin-independent pathway, autophagic receptors localize to mitochondria and interact with microtubule-associated protein 1A/1B light chain 3 to trigger mitochondrial destruction [195]. While basal mitophagy is relatively active in podocytes and necessary to maintain mitochondrial homeostasis, it is low in proximal tubular cells and enhanced in stressed cells [197,198].

Impaired mitophagy may contribute to the development and progression of DKD. Excessive accumulation of damaged mitochondria in diabetic kidney caused increased oxidative stress and inflammation [199,200]. PINK1/Parkin-mediated mitophagy is reported decreased in animal models and diabetic patients [199,201], and restoring mitophagy protected against the progression of DKD [201]. Podocyte-specific [201,202] and proximal tubule-specific [203] knockout for the autophagy related in five models of diabetes and obesity caused podocyte damage, glomerulosclerosis, and proteinuria, respectively. While the causal connection between impaired mitochondrial fitness and DKD progression has been established, more effort must be made to identify the role of bioenergetics in this complex equation.

## 6. Therapeutic Approaches to Improve Kidney Bioenergetics and DKD

The efforts to achieve normal circulating glucose levels are challenging due to the risk of hypoglycemia. This risk is amplified by the impaired kidney function that decreases the metabolism and excretion of the antidiabetic drugs, and the elimination of the kidney tubules contribution to gluconeogenesis as a source of endogenous glucose. Exceptions are represented by the glucagon-like peptide 1 (GLP-1) receptor agonists and the dipeptidyl peptidase 4 (DPP-4) inhibitors that can be used in patients with impaired renal function, whereas the use of the inhibitors of sodium-glucose cotransporter 2 (SGLT2) is limited [204]. Therefore, novel therapeutic strategies have been explored to delay the progression of DKD. This review will summarize the results of preclinical studies supporting the concept that targeting mitochondrial pathways may benefit DKD progression.

Although it is accepted that oxidative stress is a critical pathogenic mechanism for DKD, clinical trials using systemic antioxidants have conflicting outcomes and safety concerns [205,206]. Specifically targeting mitochondrial ROS must yet be proven as beneficial in human subjects while they are promising in experimental DKD models. Preclinical data with Szeto-Schiller (SS) peptides (SS-31, Elamipretide or Bendavia) showed improved mitochondrial biogenesis in rodent models of DKD [207] by maintaining cytochrome c, oxidative phosphorylation and ATP generation, inhibiting mitochondrial swelling in renal tubular epithelial cells, and mitochondrial cristae formation in experimental diabetes. Successful stabilization of cardiolipin, an inner membrane mitochondrial phospholipid required for maintaining ETC complex integrity and cristae formation, has also been shown to improve key features DN [207,208,209]. SS peptides prevented peroxidation of cardiolipin and mitigated mesangial expansion and podocyte rarefaction in db/db mice [108], and protected mitochondrial cristae morphology in proximal tubules in a rat model of acute kidney [207]. The administration of the mitochondrial-targeted coenzyme Q (MitoQ) maintained mitochondrial integrity and function in the Akita T1D mouse model while improving tubular and glomerular function, and decreasing glomerular damage, albuminuria, and interstitial fibrosis [210]. It is reported that the effectiveness of mitoQ in diabetic renal protection is nearly equal to that of angiotensin-converting enzyme inhibition [211]. MitoQ decreased mitochondrial fission in DKD [212]. The protective effects of mitoQ in DKD can be attributed to its antioxidant effect [213]. MitoTEMPO, an SOD2 mimetic mitochondrial targeted antioxidant, can alleviate key features of diabetic nephropathy [214,215].

AMP-activated kinase (AMPK) is markedly decreased in kidneys of diabetic mice and humans [216,217,218]. Administration of AICAR and AMPK activator, prevented DKD in diabetic models, at least in part, through increased mitochondrial biogenesis and ETC activity [115,219], and prevented glomerulopathy and tubulointerstitial fibrosis in mice by stimulating FA oxidation [220]. AMPK activation is proposed to treat DKD by preventing renal fibrosis, extracellular matrix accumulation, apoptosis, and inflammation [221] 

Peroxisome proliferator-activated receptors (PPARs) regulate cellular metabolism, mitochondrial biogenesis and function, FA oxidation, and glucose homeostasis. Fenofibrate activates PPARα leading to the stimulation of lipoprotein lipase and FA oxidation. Bardoxolone increases PPARγ level and mitochondrial biogenesis [222], but the use in diabetic human patients has not been beneficial [206]. In animal models of diabetes, the treatment with fenofibrate led to a decrease in hyperglycemia, insulin resistance, and FA accumulation in the kidney [223,224], and mitigated dyslipidemia and albuminuria in patients with T2DM while reducing the risk of cardiovascular events [225].

Adrenergic receptors are widely expressed in the kidney and correlate with mitochondrial function. The β2-adrenergic receptor and 5-hydroxytryptamine receptor 1F stimulate mitochondrial biogenesis and function. Formoterol, a β2 adrenergic agonist, activated the PGC1α-dependent mitochondrial biogenesis in podocytes and tubular cells in mice and contributed to the recovery of renal function [226,227,228].

Renal (Pro)renin receptor (PRR) negatively regulates mitochondrial biogenesis and function in diabetic kidney by decreasing the PGC-1α/AMPK/SIRT-1 signaling pathway. In diabetic mice, PRR expression is upregulated, leading to increased mitochondrial-generated oxidative stress that promotes DKD. PRR downregulation mitigated albuminuria and glomerular hypertrophy, and prevented the reduction in PGC-1α and the decrease in mtDNA while improving mitochondrial function. Additionally, downregulation of PRR improved the SIRT-1/AMPK expression in response to high glucose [229]. 

PGC1α masters mitochondrial dynamics and bioenergetics. High glucose downregulates PGC1α in kidney podocytes of kidneys. The administration of the plant alkaloid, berberine, reversed podocyte damage and glomerulosclerosis in diabetic mice by suppressing lipid accumulation, mitochondrial oxidative stress, and dysfunction, and correcting FA oxidation via upregulating PGC1α in podocytes [192]. PGC1α overexpression in mesangial cells promoted mitochondrial biogenesis and FA oxidation, stimulated mitochondrial translation, and inhibited kidney fibrosis and podocyte injury.

An increase in NAD pool can be achieved by supplementation with either NAD or its precursors. NAD supplementation prevented the high glucose-induced mesangial hypertrophy via activating SIRT1 and SIRT3 [176]. Administration of NMN, an intermediate in NAD synthesis, alleviated inflammation and fibrosis in T1D diabetic rats [157], and increased the oxidized form, NAD^+^, SIRT1 expression, and NAD^+^ salvage pathway in the kidney, ameliorating albuminuria, mesangial expansion, and foot process effacement in T2D mice [230]. However, the benefit of correcting the NAD pool in the human kidney is still uncertain. NAMPT, a limiting enzyme of NAD^+^ synthesis, is increased in T1D diabetic rats, and exogenous NAMPT induces kidney inflammation [157,231]. NAMPT overexpression caused inflammation and fibrosis in T1D rats through suppressing SIRT1 [157]. In addition, exogenous NR administration had no benefits in young healthy animals [232] or human subjects [233]. A randomized, double-blind, placebo-control study showed that a short-term administration of a combination of NR supplementation and a SIRT activator increased NAD^+^ levels but was not beneficial on renal function in patients with acute kidney injury [234].

SIRT activation may be achieved by caloric restriction and specific activators. Caloric restriction has been protective in animal models of kidney damage [235,236]. It is also reported that caloric restriction and intermittent fasting prevented DKD via improving NAD-dependent SIRT pathway [237], alleviated kidney inflammation in diabetic mice via activating SIRT1 [238], reduced renal oxidative stress and inflammation via activating SIRT2 [239], and attenuated palmitate-induced ROS production and inflammation in proximal tubular cells via SIRT3-mediated deacetylation [240,241].

Several SIRT-activating compounds have been identified to protect against DKD development by reducing mitochondrial oxidative stress, apoptosis, and inflammation. The SIRT1 agonists, resveratrol [242,243], and BF175 [172], reduced mitochondrial oxidative stress and apoptosis in podocytes of diabetic mice via regulating SIRT1/PGC-1α and SIRT1/p53 signaling, and the SIRT1/antioxidant response element pathway [244], renal fibrosis, and oxidative stress [245]. Some common anti-diabetic drugs that have been widely used clinically can also activate SIRTs. For example, several studies showed that metformin reduced oxidative stress and enhanced autophagy in rat mesangial cells and podocytes, and further protected against DKD progression [216,246]. SGLT2 inhibitors canagliflozin reversed high glucose-induced SIRT1 suppression [247] in human renal tubular cells and T2D mice. In addition, the SGLT2 inhibitor, empagliflozin, restored high glucose-induced SIRT3 suppression and decreased kidney fibrosis [248].

PARP is an NAD consumer. Tempol (ebselen), a mitochondrial membrane-permeable SOD-mimetic, reduced podocytes apoptosis via suppressing PARP signaling in T1D rats [249]. The specific PARP inhibitors, INO-1001 and PJ-34, suppressed high glucose-induced oxidative stress and nuclear NF-κB activation in podocytes and T2D mice, and podocytes apoptosis in T1D rats [249,250]. Another PARP inhibitor, 3-aminobenzamide, inhibited high glucose-induced oxidative stress [251,252]. Suppression of the NAD^+^-degrading enzyme, CD38, increased endogenous NAD^+^ level and restored mitochondrial function [253,254] via increasing SIRT3 activity in renal tubular cells of diabetic rats [168]. CD38 inhibition also decreased renal inflammation in T1D rats [255] and decreased glucose-induced oxidative stress, injury, and inflammation in human renal tubular epithelial cells [256].

Increased proximal sodium and chloride reabsorption via SGLT2 triggers the tubulo-glomerular feedback from the macula densa to increase glomerular pressure and filtration. SGLT2 inhibition is an approved therapeutic approach to lower glucose level and HbA1c in patients with T2D [257] via decreasing tubular reabsorption capacity and increasing urinary glucose loss. In addition, by decreasing proximal sodium and chloride reabsorption, SGLT2 inhibitors limit the diabetes-induced glomerular hyperfiltration [18] and preserve kidney function [258,259]. Interestingly, the effect of SGLT2 on glomerular filtration is independent of changes in circulating blood glucose levels [260], indicating that their benefit on kidney function is not mediated solely by providing good glucose control. SGLT2 inhibitors improved brain mitochondrial function, insulin sensitivity, and cognitive function [261]; were cardioprotective by preserving mitochondrial integrity and decreasing oxidative stress [262] when administered to obese insulin resistant rats; and increased mitochondrial biogenesis signal in adipocytes [263]. In a model of T1D, empagliflozin, a SGLT2 inhibitor, preserved mitochondrial quality control in renal tubular cells cultured in diabetic conditions [264], suggesting that the renoprotective effect of SGLT2 may be mediated by maintaining mitochondrial integrity. Future research is needed to unfold the mechanisms of how SGLTs inhibitors protect mitochondrial metabolism and decrease oxidative stress in various kidney cells in diabetes.

## 7. Conclusions and Future Directions

Diabetic kidney disease has a multifactorial pathogenesis. While hyperglycemia is considered the initiating factor responsible for glomerular hyperfiltration and increased tubular reabsorbtion, the mechanisms responsible for tubulointerstitial disease and DKD progression are elusive. Impaired mitochondrial bioenergetics, manifested as decreased mitochondrial number, integrity, and function, may lead to reductive and oxidative stress and energy deficit, thus compromising cellular structure and function and entertaining vicious processes with kidney damage.

Future research must take into account the metabolic heterogeneity of various kidney cells and elucidate their individual contribution and crosstalk to the DKD progression. Renal epithelial cells resemble cardiomyocytes and rely on FA oxidation as the major ATP provider. The presence of a full glycolytic pathway suggests that tubular cells may switch energetic substrates during physiological and pathological conditions. Research into potential alterations in the kidney metabolic flexibility in diabetes is needed to unfold its role in DKD progression.

It is currently recognized that mitochondrial defects lead to a large spectrum of abnormalities in addition to the canonical energy deficit and oxidative stress. Knowledge regarding changes in the NADH/NAD redox ratio in kidney cells and cellular compartments, including mitochondria, in diabetes is largely limited and needs further investigation. Correcting the kidney NAD pool may be a critical therapeutic approach to improve mitochondrial metabolism in the diabetic kidney. Although present in the kidney and recognized for its role in altering the cardiac NADH/NAD redox ratio, the mitochondrial enzyme nicotinamide nucleotide transhydrogenase (NNT) has not been investigated in normal and diabetic kidney cells. NNT may be a novel therapeutic target to correct mitochondrial metabolism in diabetic kidney.

## Figures and Tables

**Figure 1 cells-10-02945-f001:**
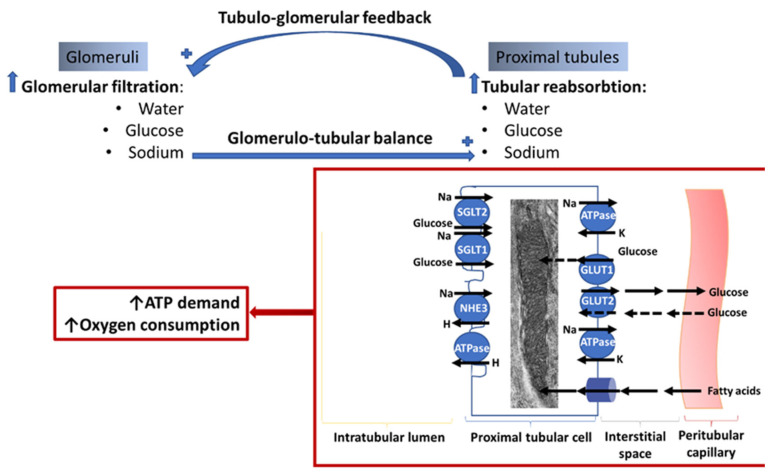
The crosstalk between glomeruli and proximal tubules is an energy-consuming process. The increased proximal tubular reabsorption of components within the glomerular filtrate involves apical transporters including Sodium Glucose Transporters (SGLT) 1 and 2, Sodium Hydrogen Exchanger 3 (NHE3), and Hydrogen-ATPase. The gradients needed for the apical transport are provided by the Na-K ATPase. Glucose exits via basal GLUT2 to enter the peritubular capillaries. Hyperglycemia either reverse or impede the glucose transport via GLUT2 and facilitate the glucose uptake via GLUT1, thus favoring intracellular glucose accumulation. The major ATP source is fatty acid (FA) β-oxidation within the mitochondria; however, glucose can also be used for energy generation although the bulk is only in transit to be recovered from the urinary space to the blood.

**Figure 2 cells-10-02945-f002:**
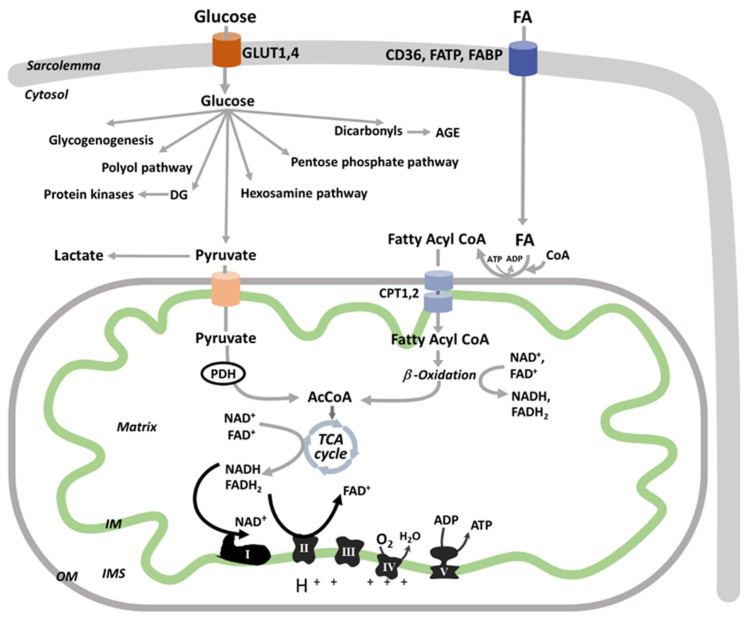
Metabolism of energetic substrates in kidney cells. The basal and insulin-dependent glucose uptake into kidney cells is mediated by the Glucose Transporters (GLUT) 1 and 4. Free fatty acids (FA) enters kidney cells by a route supported by the protein CD36, FA transport proteins (FATPs) [69], and FA binding proteins (FABPs) [70]. FA uptake by kidney cells is not hormonally regulated and driven by their circulating availability. Glucose follows multiple metabolic pathways including glycolysis, glycogen synthesis (glycogenogenesis), polyol pathway (with sorbitol and fructose formation), conversion to either diacylglycerol (DAG) to activate protein kinases or to dicarbonyls (i.e., methylglyoxal) to form advanced glycation end products (AGE), and is shuttled into the hexosamine biosynthetic or pentose phosphate pathways. Pyruvate is either converted to lactate (aerobic glycolysis [71]) or transported into mitochondria via a mitochondrial pyruvate carrier to be converted by pyruvate dehydrogenase (PDH) to acetyl-CoA (AcCoA) for the tricarboxylic acid (TCA) cycle. For simplicity, extramitochondrial glucose fluxes are not shown in detail. After entry into the cell, long chain FAs are activated to FA-CoA that is either esterified as triacylglycerol (stored in the cytosol, not shown) or enter the mitochondria via carnitine palmitoyltransferases (CPT1 and 2) to be oxidized. The end products of each FA β-oxidation cycle are NADH, FADH_2_, and acetyl-CoA, which are further oxidized by the electron transport chain (ETC) complexes or TCA, respectively, ultimately leading to ATP synthesis via mitochondrial oxidative phosphorylation. Mitochondrial oxidative phosphorylation is the main ATP provider. While electrons are transferred from the reducing equivalents, NADH and FADH2, to oxygen by the ETC complexes, an electrochemical gradient is built across the mitochondrial inner membrane (IM), which is used by the ATP synthase (complex V) to form ATP. OM: outer membrane; IMS: intermembrane space.

**Figure 3 cells-10-02945-f003:**
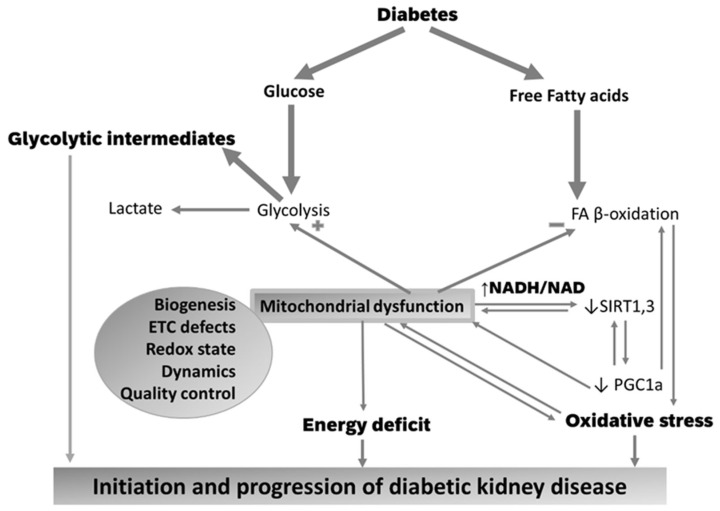
Excessive energetic substrates in diabetes leads to mitochondrial defects that cause reductive stress, oxidative stress, and energy deficit. Decreased insulin activity in diabetes provides an excess of circulating energetic substrates (i.e., glucose, free fatty acids) that enter the kidney cells by an unregulated route and are used in metabolism. An increase in the glycolytic flux that is not matched by an active mitochondrial oxidative phosphorylation leads to an accumulation of glycolytic intermediates that follow alternative metabolic pathway in the cytosol, which are responsible for inflammation, fibrosis, and epigenetic changes. The increased fatty acid (FA) oxidation observed in early diabetes, which is driven by PGC1α overexpression, leads to increased mitochondrial oxidative stress. During more advanced diabetes, mitochondrial defects limit FA oxidation and increase the NADH/NAD redox ratio (reductive stress) that restrains the mitochondrial lysine deacetylase, sirtuin 3 (SIRT3) activity. SIRT3 deacetylates and activates FA oxidation enzymes, ETC components, and PGC1α. Mitochondrial dysfunction has multiple pathogenic mechanisms including decreased PGC1α -mediated mitochondrial biogenesis and posttranslational modifications of mitochondrial proteins (oxidation, lysine acetylation).

**Figure 4 cells-10-02945-f004:**
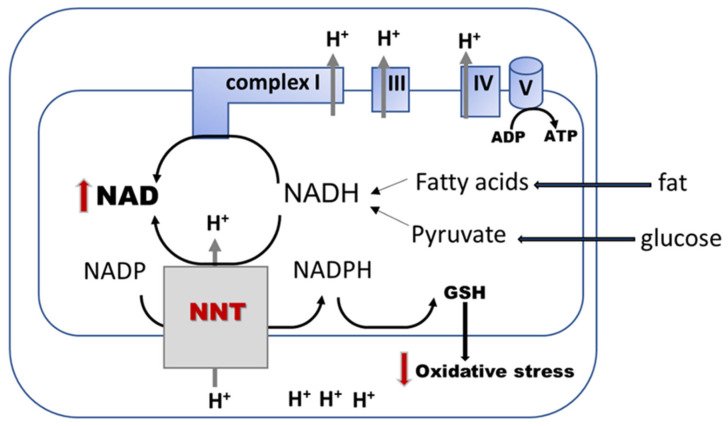
The role of nicotinamide nucleotide transhydrogenase (NNT) in decreasing the reductive and oxidative stress. NNT is an inner mitochondrial membrane enzyme that transfers electrons from mitochondrial NADH to NADPH by using the inner membrane proton motive force to boost the peroxide antioxidant defense, such as reduced glutathione (GSH) [186,187]. The enzyme reduces NADPH by oxidizing NADH that is generated by oxidizing energetic substrates (i.e., fat and glucose), thus decreasing the reductive stress (NADH/NAD ratio).

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
