# Peer review of "Mitochondria in Diabetic Kidney Disease"

_cells, 2021, doi:10.3390/cells10112945_

Round 1
Reviewer 1 Report
I read with pleasere this review about mythocondrial dysfunction in diabetic kidney disease. The manuscript is well written and covers all the relevant aspects of the topic.
Minor comments:
- A recent review about DKD deals with those forms not characterized by proteinuria (10.3390/ijms22115425), the Authors may want to consider adding it among the references.
- I cannot see the inset (electron micrograph) in figure 2.
- Figure 2, line 248: "... to activate protein kinases or to dicarbonyls (i.e., methylglyoxal) to for advanced glycation end products (AGE)" please correct.
- What do the Authors mean with "It is conceivable that with the progression of diabetes and DKD, the NADH/NAD+ ratio would also increase with the severity of DKD and that recover upon DKD remission"? What do the Authors consider as DKD remission? Kidney transplant? Please clarify or rephrase.
- Line 802: "ROS must yet be prove as beneficial" please correct as "ROS must yet be proven as beneficial".
- The revision of potentially useful drug in the pre-clinical stage is very interesting. However, I would suggest discussing further the antioxidant role of SGLT2 inhibitors, which are the most promising diabetic kidney disease protecting drugs already available on the market (which use is allowed in CKD patients up to 45 ml/min of glomerular filtration rate).
Author Response
Review 1
I read with pleasure this review about mitochondrial dysfunction in diabetic kidney disease. The manuscript is well written and covers all the relevant aspects of the topic.
We thank the reviewer for the positive comments about our work.
Minor comments:
- A recent review about DKD deals with those forms not characterized by proteinuria (10.3390/ijms22115425), the Authors may want to consider adding it among the references.
Thank you. We added this citation in the bibliography.
- I cannot see the inset (electron micrograph) in figure 2.
We eliminated the inset in Figure 2, and apologize for the error in the Figure legend.
- Figure 2, line 248: "... to activate protein kinases or to dicarbonyls (i.e., methylglyoxal) to for advanced glycation end products (AGE)" please correct.
We corrected this mistake.
- What do the Authors mean with "It is conceivable that with the progression of diabetes and DKD, the NADH/NAD+ ratio would also increase with the severity of DKD and that recover upon DKD remission"? What do the Authors consider as DKD remission? Kidney transplant? Please clarify or rephrase.
We eliminated this sentence, and rephrase the following sentence to:
“The oversupply of energy nutrients has the potential to increase the NADH/NAD+ redox ratio in parallel with the severity of DKD. “
- Line 802: "ROS must yet be prove as beneficial" please correct as "ROS must yet be proven as beneficial".
We corrected to: “ROS must yet be proven as beneficial.”
- The revision of potentially useful drug in the pre-clinical stage is very interesting. However, I would suggest discussing further the antioxidant role of SGLT2 inhibitors, which are the most promising diabetic kidney disease protecting drugs already available on the market (which use is allowed in CKD patients up to 45 ml/min of glomerular filtration rate).
We added the following paragraph at the end of Section 6:
Increased proximal sodium and chloride reabsorption via SGLT2 triggers the tubulo-glomerular feedback from the macula densa to increase glomerular pressure and filtration. SGLT2 inhibition is an approved therapeutic approach to lower glucose level and HbA1c in patients with T2D [1] via decreasing tubular reabsorption capacity and increasing urinary glucose loss. In addition, by decreasing proximal sodium and chloride reabsorption, SGLT2 inhibitors limit the diabetes-induced glomerular hyperfiltration [2] and preserve kidney function [3,4]. Interestingly, the effect of SGLT2 on glomerular filtration is independent on changes in circulating blood glucose levels [5] indicating that their benefit on kidney function is not mediated solely by providing a good glucose control. SGLT2 inhibitors improved brain mitochondrial function, insulin sensitivity and cognitive function [6], were cardioprotective by preserving mitochondrial integrity and decreasing oxidative stress [7] when administered to obese insulin resistant rats, and increased mitochondrial biogenesis signal in adipocytes [8]. In a model of T1D, empagliflozin, a SGLT2 inhibitor, preserved mitochondrial quality control in renal tubular cells cultured in diabetic conditions [9], suggesting that the renoprotective effect of SGLT2 may be mediated by maintaining mitochondrial integrity. Future research is needed to unfold the mechanisms of how SGLTs inhibitors protect mitochondrial metabolism and decrease oxidative stress in various kidney cells in diabetes.
- Vallon, V.; Thomson, S.C. Targeting renal glucose reabsorption to treat hyperglycaemia: the pleiotropic effects of SGLT2 inhibition. Diabetologia 2017, 60, 215-225, doi:10.1007/s00125-016-4157-3.
- Vallon, V.; Thomson, S.C. The tubular hypothesis of nephron filtration and diabetic kidney disease. Nat Rev Nephrol 2020, 16, 317-336, doi:10.1038/s41581-020-0256-y.
- Wanner, C.; Inzucchi, S.E.; Lachin, J.M.; Fitchett, D.; von Eynatten, M.; Mattheus, M.; Johansen, O.E.; Woerle, H.J.; Broedl, U.C.; Zinman, B., et al. Empagliflozin and Progression of Kidney Disease in Type 2 Diabetes. N Engl J Med 2016, 375, 323-334, doi:10.1056/NEJMoa1515920.
- Zelniker, T.A.; Wiviott, S.D.; Raz, I.; Sabatine, M.S. SGLT-2 inhibitors for people with type 2 diabetes - Authors' reply. Lancet 2019, 394, 560-561, doi:10.1016/S0140-6736(19)30699-3.
- Heerspink, H.J.; Desai, M.; Jardine, M.; Balis, D.; Meininger, G.; Perkovic, V. Canagliflozin Slows Progression of Renal Function Decline Independently of Glycemic Effects. J Am Soc Nephrol 2017, 28, 368-375, doi:10.1681/ASN.2016030278.
- Sa-Nguanmoo, P.; Tanajak, P.; Kerdphoo, S.; Jaiwongkam, T.; Pratchayasakul, W.; Chattipakorn, N.; Chattipakorn, S.C. SGLT2-inhibitor and DPP-4 inhibitor improve brain function via attenuating mitochondrial dysfunction, insulin resistance, inflammation, and apoptosis in HFD-induced obese rats. Toxicol Appl Pharmacol 2017, 333, 43-50, doi:10.1016/j.taap.2017.08.005.
- Durak, A.; Olgar, Y.; Degirmenci, S.; Akkus, E.; Tuncay, E.; Turan, B. A SGLT2 inhibitor dapagliflozin suppresses prolonged ventricular-repolarization through augmentation of mitochondrial function in insulin-resistant metabolic syndrome rats. Cardiovasc Diabetol 2018, 17, 144, doi:10.1186/s12933-018-0790-0.
- Yang, X.; Liu, Q.; Li, Y.; Tang, Q.; Wu, T.; Chen, L.; Pu, S.; Zhao, Y.; Zhang, G.; Huang, C., et al. The diabetes medication canagliflozin promotes mitochondrial remodelling of adipocyte via the AMPK-Sirt1-Pgc-1alpha signalling pathway. Adipocyte 2020, 9, 484-494, doi:10.1080/21623945.2020.1807850.
- Lee, Y.H.; Kim, S.H.; Kang, J.M.; Heo, J.H.; Kim, D.J.; Park, S.H.; Sung, M.; Kim, J.; Oh, J.; Yang, D.H., et al. Empagliflozin attenuates diabetic tubulopathy by improving mitochondrial fragmentation and autophagy. Am J Physiol Renal Physiol 2019, 317, F767-F780, doi:10.1152/ajprenal.00565.2018.

Reviewer 2 Report
This is a well written and comprehensive review that covers an important topic. The review highlights mitochondrial abnormalities in the diabetic kidney and outlines all aspects of diabetes affecting bioenergy metabolism and also summarizes the results of preclinical studies supporting the concept that targeting mitochondrial pathways may benefit DKD progression. This review nicely summarizes the role of mitochondria in DKD, which has certain guiding value for studying mitochondria in diabetic nephropathy. I have some comments to the manuscript.
- The abstract should be an unstructured and concise synopsis. Introduction in abstract section is too long. There is an unusual amount of detail in this section mentioning the epidemiology, clinical manifestations and pathogenesis of DKD. This is unnecessary in the context of the entire review and the authors are encouraged to shorten this section.
- The pictures shown in this paper are too vague, I suggest the author redraw them.
- The number of the subtitle in the article is confusing. For example, the number on line 219 and line 295, line 488 and 560 are the same.
- Authors put too much emphasis on the summarizing major findings of the present research, but I fear these are getting lost in personal opinion. In the summary part, the author should put forward more personal opinions about future development direction, which should be able to inspire readers to achieve more new theories in this area.
Author Response
Review 2
This is a well written and comprehensive review that covers an important topic. The review highlights mitochondrial abnormalities in the diabetic kidney and outlines all aspects of diabetes affecting bioenergy metabolism and also summarizes the results of preclinical studies supporting the concept that targeting mitochondrial pathways may benefit DKD progression. This review nicely summarizes the role of mitochondria in DKD, which has certain guiding value for studying mitochondria in diabetic nephropathy. I have some comments to the manuscript.
We thank the reviewer for the positive comments about our work.
- The abstract should be an unstructured and concise synopsis. Introduction in abstract section is too long. There is an unusual amount of detail in this section mentioning the epidemiology, clinical manifestations and pathogenesis of DKD. This is unnecessary in the context of the entire review and the authors are encouraged to shorten this section.
We have shorted the abstract as suggested by the reviewer.
Diabetic kidney disease (DKD) is the leading cause of end stage renal disease (ESRD) in the US. The pathogenesis of DKD is multifactorial and involves activation of multiple signaling pathways with merging outcomes including thickening of the basement membrane, podocyte loss, mesangial expansion, tubular atrophy, and interstitial inflammation and fibrosis. The glomerulo-tubular balance and tubule-glomerular feedback support an increased glomerular filtration and tubular reabsorption, with the latter relying heavily on ATP and increasing the energy demand. There is evidence that alterations in mitochondrial bioenergetics in kidney cells lead to these pathologic changes and contribute to the progression of DKD towards ESRD. This review will focus on the dialogue between alterations in bioenergetics in glomerular and tubular cells and its role in the development of DKD. Alterations in energy substrate selection, electron transport chain, ATP generation, oxidative stress, redox status, protein posttranslational modifications, mitochondrial dynamics and quality control will be discussed. Understanding the role of bioenergetics in the progression of diabetic DKD may provide novel therapeutic approaches to delay its progression to ESRD.
- The pictures shown in this paper are too vague, I suggest the author redraw them.
We have redrawn Figures 1 and 3, and tried to make them more relevant to our discussion.
- The number of the subtitle in the article is confusing. For example, the number on line 219 and line 295, line 488 and 560 are the same.
We corrected the number of subtitles as suggested by the reviewer.
- Authors put too much emphasis on the summarizing major findings of the present research, but I fear these are getting lost in personal opinion. In the summary part, the author should put forward more personal opinions about future development direction, which should be able to inspire readers to achieve more new theories in this area.
We agree with the reviewer, and have added an additional paragraph addressing future directions in Section 7.
Renal epithelial cells resemble cardiomyocytes and rely on FA oxidation as the major ATP provider. The presence of a full glycolytic pathway suggest that tubular cells may switch energetic substrates during physiological and pathological conditions. Research into potential alterations in the kidney metabolic flexibility in diabetes is needed to unfold its role in DKD progression.
It is currently recognized that mitochondrial defects lead to a large spectrum of abnormalities in addition to the canonical energy deficit and oxidative stress. Knowledge regarding changes in the NADH/NAD redox ratio in kidney cells and cellular compartments, including mitochondria, in diabetes is largely limited and needs further investigation. Correcting the kidney NAD pool may be a critical therapeutic approach to improve mitochondrial metabolism in the diabetic kidney. Although present in the kidney and recognized for its role in altering the cardiac NADH/NAD redox ratio, the mitochondrial enzyme nicotinamide nucleotide transhydrogenase (NNT) has been not investigated in normal and diabetic kidney cells. NNT may be a novel therapeutic target to correct mitochondrial metabolism in the diabetic kidney.
